# An Adaptive TTT Handover (ATH) Mechanism for Dual Connectivity (5G mmWave—LTE Advanced) during Unpredictable Wireless Channel Behavior

**DOI:** 10.3390/s23094357

**Published:** 2023-04-28

**Authors:** Vigneswara Rao Gannapathy, Rosdiadee Nordin, Asma Abu-Samah, Nor Fadzilah Abdullah, Mahamod Ismail

**Affiliations:** 1Department of Electrical, Electronic and Systems Engineering, Faculty of Engineering and Built Environment, Universiti Kebangsaan Malaysia, Bangi 43600, Malaysia; vigneswara@utem.edu.my (V.R.G.); adee@ukm.edu.my (R.N.); fadzilah.abdullah@ukm.edu.my (N.F.A.); mahamod@ukm.edu.my (M.I.); 2Department of Electronics and Computer Engineering Technology, Faculty of Electrical and Electronic Engineering Technology, Universiti Teknikal Malaysia Melaka, Durian Tunggal 76100, Malaysia

**Keywords:** 5G mmWave, dual-connectivity, handover, handover margin (HOM), latency, multi-RAT, Network Simulator 3 (NS3), Time-to-Trigger (TTT), throughput

## Abstract

Fifth Generation (5G) signals using the millimeter wave (mmWave) spectrums are highly vulnerable to blockage due to rapid variations in channel link quality. This can cause the devices or User Equipment (UE) to suffer from connection failure. In a dual connectivity (DC) network, the channel’s intermittency issues were partially solved by maintaining the UE’s connectivity to primary (LTE advanced stations) and secondary (5G mmWave stations) simultaneously. Even though the dual-connected network performs excellently in maintaining connectivity, its performance drops significantly due to the inefficient handover from one 5G mmWave station to another. The situation worsens when UE travels a long distance in a highly dense obstacle environment, which requires multiple ineffective handovers that eventually lead to performance degradation. This research aimed to propose an Adaptive TTT Handover (ATH) mechanism that deals with unpredictable 5G mmWave wireless channel behaviors that are highly intermittent. An adaptive algorithm was developed to automatically adjust the handover control parameters, such as Time-to-Trigger (TTT), based on the current state of channel condition measured by the Signal-to-Interference-Noise Ratio (SINR). The developed algorithm was tested under a 5G mmWave statistical channel model to represent a time-varying channel matrix that includes fading and the Doppler effect. The performance of the proposed handover mechanism was analyzed and evaluated in terms of handover probability, latency, and throughput by using the Network Simulator 3 tool. The comparative simulation result shows that the proposed adaptive handover mechanism performs excellently compared to conventional handovers and other enhancement techniques.

## 1. Introduction

The advancement of communication technology and its services in many areas of life has continuously contributed to a huge increase in the number of consumers, and in addition, it needs a greater capacity to handle more users [1]. However, this is unable to meet because data transfer speed has now risen 1000 times more than it was during previous generations (1G). The Fifth-Generation Technology (5G) and beyond is an emerging technology across the globe that has a powerful capability of supporting extremely high-speed communication requirements for several applications [2,3]. It can also handle a large gigabit of data with the utmost lower latency and has the potential to satisfy certain services that require a strong bandwidth [1,2,3].

In 2022, Shahen Shah [4] presented the evolution of cellular communication in regard to some of emerging technologies, such as MIMO, software-defined networking, and millimeter waves (mmWave). The author also gave an overview of multiple access techniques and challenges that can be anticipated in forthcoming communication. As the 5G mmWave network is very susceptible to blockage and prone to obstructions, finding ways to enable additional transmission flexibility and improve user connectivity is an important research direction in this area. In [5], the author described the use of massive MIMO systems in 5G mmWave technology to achieve multi-gigabit data rates and low-latency communication.

The signal at Beyond 5G mmWave networks works at higher frequencies (30–300 GHz), as their wavelength is in the order of millimeters. One of the fundamental issues with 5G mmWave bands is channel intermittency, where the signal condition will be varied dramatically with respect to the mobility of the User Equipment (UE) [6].

The author of [6] demonstrated the reduction in the power density of mmWave bands as they propagate through space for three outdoor scenarios (Urban Micro, Urban Macro, and Rural Macro). The path-loss study revealed that the signals in mmWave bands deteriorate when the UE moves away from the base station in the presence of obstacles. Moreover, the frequencies at these bands find it difficult to penetrate buildings and are susceptible to blockage. Even a minor obstruction, such as stone, concrete, and/or a human body, could cause prominent attenuation in their signal strength [6,7,8,9].

Therefore, mobility management is essential to ensure the smooth and uninterrupted connection of users (i.e., UEs) at these intermittent bands. The conventional mobility management solutions that were developed for LTE and LTE-A remained incompetent to handle seamless mobility issues for every handover. Moreover, the importance of mobility management is significantly increased, especially in 5G mmWave, due to its requirement for connectivity with high reliability and bounded latency [1].

Handover is one of the most important components in mobility management to ensure that user UEs move freely and change their connection to a new serving cell seamlessly, without any interruption. The intermittency issues in the mmWave channel further complicate the handover decisions, eventually affecting the network performance in terms of user connectivity, latency, and throughput. Although handover management in conventional cellular networks is relatively mature, research in this handover management field has just started for Beyond 5G cellular networks. Many researchers are enthusiastically investigating and designing a robust handover mechanism at 5G mmWave frequency bands [10,11,12,13,14,15,16,17,18,19,20,21,22,23,24,25,26,27,28,29,30,31,32].

Handovers in heterogeneous networks (HetNets) have received interest in recent years, especially when 5G ultra-dense small cells coexist with existing LTE networks. This approach is also known as DC, in which UEs in the network preserve and maintain the connection with different Radio Access Technologies (RATs), such as 5G mmWave and LTE-advanced, so that a single-link failure can be resolved by switching to alternate paths [6]. Even though channel intermittency issues were partially solved by seamlessly maintaining the connection between both RATs, the performance dropped significantly due to inefficient handover from one 5G mmWave station to another. The situation worsens when the UE travels a long distance in a highly dense obstacle environment, which requires multiple ineffective handovers that eventually lead to performance degradation. Thus, developing an efficient and robust handover technique that can provide a seamless and high-quality connection to UEs is a critical challenge that needs to be addressed in HetNets.

The main challenges encountered are namely due to (1) handling the handover/path switch in the event of rapid fluctuation of signal, (2) requiring severe scanning of channel quality, (3) consuming more time, and (4) causing high latency. In addition, the previous study revealed that unnecessary handover, even for fixed UEs, is the major drawback for higher frequency networks [10]. In addition, a lack of dynamic configuration at the handover margin (HOM) and TTT parameters often leads to inefficient handovers, which result in higher signaling overheads and performance degradation [10,11,12,13,14,15,16,17,18,19,20,21,22,23,24,25,26,27,28,29,30,31,32].

To solve these challenges, we aimed to develop an adaptive handover mechanism that automatically regulates Handover Control Parameters (HCPs), such as TTT, in this work. The contributions of the paper can be summarized as follows:The development of a novel ATH mechanism that automatically adjusts the TTT handover parameter based on the instantaneous channel conditions. The proposed ATH mechanism is an extension from previous works [19,21,27]; it exploits the UE’s information, such as SINR, to adapt the TTT as the system outputs.A performance evaluation of the proposed adaptive handover mechanism is carried out in Non-Stand Alone (NSA) DC network architecture (5G mmWave—LTE Advanced). The adaptive handover algorithm is validated under realistic mmWave channel behavior.An evaluation of the system performance in terms of handover probability, latency, and throughput with different mobility speeds and beamforming scenarios. The performance of the proposed handover mechanism is further compared with the conventional handover method and another two competitive handovers proposed in [21,23].

The rest of the paper is structured as follows: Section 2 presents related works, and then Section 3 introduces the proposed adaptive TTT handover mechanism. Section 4 describes the simulation framework and performance metrics, while Section 5 presents the performance analysis and results. Finally, in the last section, we conclude our paper.

## 2. Related Works

This section gives an overview of the relevant handover mechanisms/solutions that proposed to be implemented in the 5G technology. Table 1 presents a summary of most recent related works.

Alraih et al. [15] proposed a Robust Handover Optimization Technique with a Fuzzy Logic Controller known as RHOT-FLC. In this work, the author proposed a technique to automatically configure HCPs’ setting (TTT and HOM) by utilizing information such as the Reference Signal Received Power (RSRP), Reference Signal Received Quality (RSRQ), and UE velocity. RHOT-FLC exploits fuzzy logic techniques in this work, and it needs to go through 5 different stages (input, fuzzification, inference engine, defuzzification, and output) and apply 48 rules before dynamically estimates TTT and HOM for each process. MATLAB was used to simulate and validate the efficiency of the proposed scheme, and according to the findings, the author demonstrated some improvements over commonly used handover metrics, such as Handover Probability (HOP), Handover Failure (HOF), Handover Ping Pong (HOPP), and Handover Interruption Time (HIT) performances. Although fuzzy-logics-based RHOT-FLC systems are able to achieve a great handover performance, the time required to perform optimization is lengthy, and it increases as the input parameters and rules increase.

In 2022, Hwang et al. [16] proposed a novel adaptive handover scheme that is known as the fuzzy-logic-based handover algorithm with dynamic HOM and TTT (FLDHDT) to dynamically adjust the HCPs, such as TTT and HOM, by exploiting two important parameters, i.e., the SINR and moving speed of UE as an input to fuzzy logic (FL) controller. The author considered the NSA networking mode with the coexistence of 5G and 4G in this work. The proposed scheme requires two stages, namely data preprocessing and inference using FL, before make handover decision with an appropriate setting of TTT and HOM. The evaluation was carried out by using NS-3 tool, and the author proved that the performance of the proposed method greatly improved in terms of the number of handovers, overall system throughput, and HOPP ratio. However, the study was only focused on handover (HO) events, HOPP, and system throughput and did not discuss other important HO KPIs, such as latency, Radio Link Failure (RLF), HIT, and cell edge Spectral Efficiency (SE).

In [17], Karmakar proposed a novel smart handover method to intelligently adapt TTT and hysteresis values. The proposed method, known as Learning-Based Intelligent Mobility Management (LIM2), was designed by using a Kalman filter that consists of three stages: (i) predict signal quality of the serving and neighbor base stations, (ii) select the target base station by using reinforcement learning method, and (iii) adapt the TTT and hysteresis by using the greedy policy. The author conducted an extensive analysis by using the NS3 simulation tool and proved that LIM2 significantly improves the handover performance in different mobility scenarios in terms of throughput, packet loss rate, and user connectivity. However, this method is catered toward different mobility scenarios, without taking other parameters, such as channel intermittency, blocking events, and other mmWave channel behaviors, into consideration.

In Alhammadi et al.’s paper [18], the authors proposed a speed-based self-optimization algorithm known as the Auto-Tuning Optimization (ATO), which automatically enables the system to modify the HCPs in LTE-Advanced/5G networks. The proposed algorithm uses the user’s received velocity and power as input for the system to modify the TTT and HOM automatically. The comparative results revealed that the suggested handover significantly outperformed the other competitive algorithms. However, the algorithms must be developed with high interoperability and validated using Beyond 5G networks.

Shayea et al. [19] proposed a weighted-function-based handover optimization technique. The proposed algorithm, known as Adaptive Handover Margin based on Novel Weight Function (AHOM-NWF), automatically modifies HCPs such as HOM depending on three different parameters as the system’s input: user’s velocity, traffic, and SINR. The simulation results revealed that the suggested weighted-function-based optimization technique improved the spectral efficiency (SE) at the edge of the cells. However, the algorithm in this work only catered to different UEs speeds, traffic, and SINR, without considering the network environment with the presence of obstacles.

In [20], the author investigated the handover performance in 5G small cells network with different metrics and by considering a realistic urban channel model. The technique known as Individualistic Dynamic Handover Parameter Optimization Algorithm Based on an Automatic Weight Function (IDHPO-AWF) was proposed in this work. The author performed a comparative analysis of traditional RSSI- and SNR-based handover schemes. The Nakagami and Free Path Loss channel models were used in this study to observe the handover performance in terms of throughput, average packet delays, bit error rate, and packet loss rates. The author also concluded that the Nakagami channel model gave an accurate result and that it is suitable to study the handover performance of realistic channel environments with obstacles.

Handover failure in a 5G mmWave network is a major problem, and it can be prevented by modifying HCPs such as TTT and HOM. The authors in [21] proposed a dynamic handover control parameter algorithm to dynamically adjust HCPs according to various user mobility and speed scenarios. The results show that the proposed algorithm greatly decreases the likelihood of handovers, ping-pong effects, and radio connection failures and enhances network stability. The algorithm in this work only catered to different UEs’ speeds, without taking other parameters, such as channel intermittent behavior, into consideration.

Murtaza Cicioglu [22] proposed an entropy-based simple additive weighting decision-making method to solve some of handover issues, such as latency, handover failures, frequent handover, and the ping-pong effect, that are caused by the inefficient placement of a small cell in dense or ultra-dense 5G networks. In this work, the author used bandwidth, user density, and SINR parameters to provide a robust connection between the UE and base station. Through the finding, the author proved that the proposed approach has achieved 15% improvement over handover delay, 48% improvement over blocking probability, and 22% improvement over throughput compared to the conventional LTE handover. However, the study did not consider other aspects that may impact UE and the network during handover, such as TTT and HOM, cell edge SE, packet loss, HIT, and HOPP.

Juwon Kim et al. [23] proposed an adaptive handover scheme to implement in the LTE network and examined it for a variety of user velocities. The HCP settings such as TTT was adaptively adjusted by exploiting UE location based on the received signal strength (RSS). The OPNET simulation tool is used to validate the efficiency of the proposed scheme, and the author proved that the link failure rate and ping-pong effect are improved over the traditional handover algorithm. However, the algorithm was not designed to support different user mobility and wireless channel behaviors. Furthermore, not all handover KPIs were investigated in this work to prove the efficiency of the proposed algorithm.

Giordani et al. [24] presented a new method to select an optimal cell to perform handover from serving to the target node. The authors identified the major challenge in tracking the direction of each angular space, along with its power and timing. They proposed a unique technique to measure link quality by (i) transmitting a sounding reference signal (SRS) in directions that cover the whole angular space, (ii) collecting and gathering instantaneous signal strength, and (iii) guiding the handover and scheduling decisions. The proposed technique performs greatly in selecting an optimal mmWave cell for handover purposes under certain conditions, as highlighted in the paper. This solution also can potentially increase the network capacity in next-generation cellular systems. However, this work does not emphasize handover decision-making parameters such as TTT and HOM.

Khosravi et al. [25] developed a beamforming method to overcome frequent handover and channel complexity issues. The reinforcement learning method is used to identify the choices of backup stations in various locations of mobile users. The obtained results in this work provide a promising solution to achieve a trajectory and instantaneous target rate. However, an accurate mobility prediction scheme that combines beamforming and handover approaches remains unsolved and left for future work.

Matalatala et al. [26] presented a method to fulfill the demand for data-intensive applications. The authors investigated the use of different beamforming architectures (i.e., digital hybrid and analog) and measured their performances in 5G cellular networks. The author concluded that the 5G networks need 15% more base stations with four times less power to provide more capacity to the users and the same coverage performances in comparison with the 4G reference network. The authors of this work proposed that the digital and hybrid beamforming is the best to be implemented in Beyond 5G cellular networks.

Polese et al. [27,28] developed a novel communication protocol that allows mobile users to simultaneously establish physical layer links to 5G and LTE cells. As mmWave signals are highly susceptible to blockage and extremely intermittent, the authors presented a fast path-switching method with the help of a central coordinator through a novel uplink control signal in this work. In the event of link failure, the proposed method will react immediately to switch to the next potential and reliable link. Furthermore, the author incorporated an initial access technique proposed by Giordani in [29]. The analysis showed a substantial advantage of the proposed approach under multiple parameters compared to traditional handover methods. However, the work concentrated only on semi-statistical channel models without temporally correlated mmWave channel measurements.

Kumar et al. [30] presented two main methods, known as clustering and classifying, to build the 5G handover technique. Clustering is a process of grouping the datasets into separate units, and classification is a process of classifying user’s clustered datasets into a common path, using prediction and forecasting. The author used the k-means method for clustering and the Random Forest algorithm method for classification. By using the proposed algorithms, the predicted and forecasted datasets will be stored in the cloud. Cloud technology is used in this work as a platform for developing datasets associated with the internet. The authors used all essential ML techniques in this work, as they are easy to adapt to 5G technology. In the future, with the help of prediction and forecasting, handover management will be easier than ever, and it can be validated using other ML techniques.

Recently, research on handover studies has been directed toward the use of ML techniques. This was performed by [31], in which the Neural Network (NN) ML algorithm predicts and forecasts user’s paths while establishing handover. This approach, known as Data Driven Handover Optimization (DHO), uses a new training method to estimate the Key Performance Indexes (KPIs). The outcome of this work reveals its effectiveness in mitigating mobility problems. Massive training is required to estimate the KPI function, which is very difficult to obtain.

Despite the many propositions, the abovementioned handover algorithms were found to be incompetent in selecting an optimal HCPs setting, such as TTT and HOM, in the 5G mmWave system. Although the algorithms improved the handover performance in the proposed simulation environment and scenarios, their effectiveness dropped significantly when implemented in a 5G mmWave system that is highly intermittent. Furthermore, most of these algorithms were developed and simulated under ideal channel conditions, which do not consider mmWave channel intermittent and blocking events. Therefore, the algorithms developed under this condition may not be efficient enough to be implemented in 5G mmWave networks, and detailed investigations are required with various mobility and deployment scenarios.

## 3. Proposed Adaptive TTT Handover (ATH) Mechanism

The mmWave frequencies have very high isotropic path loss, leading to a smaller coverage area than existing networks. As a connection, the establishment of the mmWave is challenging, and profound coverage is hard to achieve. Therefore, the mmWave needs to engage with previous-generation LTE technology. A network architecture that combines the LTE and mmWave Radio Access Technologies in a fast and seamless way is important to provide ultra-reliable services for mobile users. Potential alternatives to integrating LTE and mmWave, known as DC, were proposed in [27,28]. This integration is also expanding the coverage area of the mmWave network. We used the DC framework suggested in [27] to integrate and validate our proposed solution in this work.

The important HCPs that influence the handover decisions in the networks are TTT and HOM. Both of these parameters are used to decide on an appropriate time or slot to initiate the handover to a respective target base station after fulfilment of the requirements. HOM is defined as a threshold difference between the signal strength of the serving and target base stations whereby the TTT is defined as a critical time interval that is required to meet HOM conditions.

Adapting HCPs is one of the main solutions for improving the performance of 5G mmWave networks. If the HCPs are fixed, then the UE’s connection with base stations will be unfavorably influenced, particularly when UEs differ in regard to moving speed and their changing network environment with the presence of obstacles. Hence, HCP settings must be resourcefully adapted to resolve this deficiency.

In this section, we describe the proposed novel handover mechanism called Adaptive TTT Handover (ATH) to address handover issues based on different mobility scenarios. We identified causes that disrupt the link between the UE and eNB base stations, such as early handover, late handover, and wrong cell handover. Due to the unpredictable nature of wireless channel behaviors and mobility issues in 5G mmWave, maintaining a stable connection of UE in a network is a crucial challenge. Therefore, HCPs (TTT and HOM) need to be intelligently adjusted in order to provide and optimal HO performance. The inappropriate adjustment or setting of HCPs may lead to an increase of HOP, HOF, RLF, and HOPP probability that eventually degrades the overall system performance.

The ATH mechanism is designed to adaptively adjust the HCPs (TTT), hence effectively automating the HO decision to mitigate early handover, late handover, and wrong cell handover, as mentioned earlier. As depicted in Figure 1, the proposed adaptive handover algorithm consists of four main phases: Phase 1—channel measurement phase; Phase 2—SINR evaluation phase; Phase 3—HCPs—TTT adaption and initialization phase; and Phase 4—handover execution phase. During the first phase, the system measures the signal strength of the serving and target base stations in terms of SINR. The signal strength between the serving and target base station will be extensively evaluated and compared during the second phase. In the third phase, TTT intervals will be adapted automatically based on inputs received from the first phase. At this phase, the system will continuously monitor the channel condition based on the input received from Phase 1, and if it fulfills the given requirements proposed in the ATH algorithm, then it will eventually make HO decisions. During Phase 4, the handover will be performed and executed

Table 2 depicts the different values of RSSI (i.e., RSRP, RSRQ, and SINR) parameters, which correspond to very poor (Cell Edge), poor (Mid Cell), good (Good), and very good (Excellent) signal quality. The use of RSRP and RSRQ in addition to SINR may improve the accuracy in selecting the HCPs such as TTT and HOM. However, it may increase the processing time that is crucial in the 5G mmWave network. Therefore, SINR-based uplink channel measurement reporting was selected to compromise the system performance, accuracy, and processing time.

The ATH algorithm aims to adjust TTT interval based on the instantaneous channel conditions. The proposed ATH function will estimate the appropriate TTT for each UE in the network based on the received SINR signal strength. Furthermore, it significantly improves the UEs’ connectivity in the network.

Figure 2 shows the designed framework of the proposed ATH mechanism. During the initialization stage, the maximum and minimum parameters for TTT and HOM are defined and initiated as per the 3GPP standard [33].

### 3.1. Phase 1: Channel Measurement

In this work, we considered the adaptation of a handover based on SINR signal strength, as it is a feasible option to encounter mmWave links that are interference-limited for a dense topology. In this work the SINR signal strength was computed by using the following equation:(1)$$SINR_{j,UE}=\frac{\left(\frac{P_{TX,jj}}{PL_{j,UE}}\right)\;\left(G_{j,UE}\right)}{\sum_{k\neq j}\;\left(\frac{P_{TX,kk}}{PL_{k,UE}}\right)\;\left(G_{k,UE}\right)+\left(BW\times N_{O}\right)}$$

For example, as depicted in Figure 3, the SINR between *UE*_0_ and mmWave *gNB*_1_ with the presence of mmWave *gNB*_2_ can be computed as follows:(2)$$SINR_{1,0}=\frac{\left(\frac{P_{TX,11}}{PL_{1,0}}\right)\left(\;G_{1,0}\right)}{\left(\frac{P_{TX,22}}{PL_{2,0}}\right)\;\left(G_{2,0}\right)+\left(BW\times N_{O}\right)}$$

$P_{TX,11}$ and $P_{TX,22}$ are the transmit power of mmWave *gNB*_1_ and mmWave *gNB*_2_, respectively.

$BW\times N_{O}\;$ is a thermal noise power, as defined in [6], that has a major influence on the quality of the receiver.

$PL_{1,0}$ and $PL_{2,0}$ are the path loss for mmWave *gNB*_1_ → *UE*_0_ and mmWave *gNB*_1_ → *UE*_0_, respectively. The path loss model is defined as follows:(3)$$PL_{\left(d\right)}\left[dB\right]=\alpha +\beta \;10\;log_{10}\left(D\right)$$
where *D* is the logarithmic distance that is represented by *d*/*d*_0_, where *d* is the distance between the receiver and transmitter, and *d*_0_ is the free space reference distance that is typically 1 m for the mmWave propagation model. This is a logarithmic distance path loss model where the average received signal power decreases logarithmically with distance. The parameters *α* and *β* are as defined in [6]. When a link between the UE and mmWave gNB is blocked by the obstacles, the Non-Line of Sight (NLOS) path loss state is emulated by imposing experimentally measured blockage traces [6].

The beamforming gains, $\left(G_{1,0}\right)\;\mathrm{and}\;\left(G_{2,0}\right)\;$, when the *UE*_0_ is associated with mmWave *gNB*_1_ and mmWave *gNB*_1_, respectively, can be represented as follows:(4)$$G_{1,0}=\left|w_{rx_{00}}\;H{\left(t,f\right)}_{10}\;w_{tx_{11}}\right|{}^{2}$$
(5)$$G_{2,0}={\left|w_{rx_{00}}\;H{\left(t,f\right)}_{20}\;w_{tx_{22}}\right|}^{2}$$

*H*(*t*,*f*) is a channel matrix, where *t* is the time, and *f* is the frequency.

By utilizing the SINR values, it is possible to determine the furthermost time to safely perform the handover, while avoiding other handover issues, such as RLF, HOPP, etc. In addition, to reduce the need for UE to send the measurement report back to the network, our work focused on measuring the channel quality of the uplink rather than the downlink signal. To achieve this goal, SINR-based uplink measurement reporting was considered in this work. Similar work on predictive SINR-based handover was reported in [16,19,27,32].

Furthermore, we proposed a new adaptive multicell measurement reporting system known as the next-generation Node B (gNB) measurement report table (*gMRT*) and evolved Node B (eNB) measurement report table (*eMRT*). UE directionally broadcasts a Sounding Reference Signal (SRS) in a time-varying direction to enable mmWave gNB to determine signal quality for each angular space and fill it in its *gMRT*. In this work, the signal quality, referred to by SINR, represented the highest value of each dedicated UE and mmWave gNB. The strength of SINR will be continuously scrambled by the serving gNB cell by scanning the whole angular space and direction based on different beamforming configurations. When the system uses a digital beamforming configuration, it requires 1.6 ms to collect each instance in the measurement table; meanwhile, for hybrid and analog beamforming, the system requires 12.8 ms and 25.6 ms, respectively. The cell scanning and variance can capture the channel condition and link quality. Greater detail on the beamforming configurations can be found in [27,28,29].

Once each mmWave gNBs have filled with their *gMRT*, all these reports will be sent to centralized LTE eNB through the X2 link, as illustrated in Figure 4, and subsequently built *eMRT*. Once a centralized LTE eNB station acquires full directional knowledge of signal quality from all gNBs within the network, it will adaptively tune the TTT parameter based on the proposed algorithm and subsequently make a handover decision.

Complete directional knowledge of all mmWave gNBs in the network needs to be acquired by the centralized LTE eNB before making handover decisions. Then, the process will move to the next stage of executing potential handovers in the network.

### 3.2. Phase 2: SINR Evaluation by Centralized Macro LTE eNB

Based on the proposed adaptive TTT algorithm, which is depicted in Figure 2 (Column 2), at each instant, the centralized macro-LTE base station will monitor the SINR entries in the *eMRT* and make a comparison between the following:(i)The target and serving mmWave gNB,(ii)The target and other neighboring mmWave gNB,(iii)The serving and neighboring mmWave gNB.

### 3.3. Phases 3 and 4: HCPs’ Adaptation, Initialization, and HO Execution

The handover decision (Figure 2, Column 3) is made based on the following conditions (if–then rules):(i)If all mmWave gNBs are outage with minimum SINR threshold value (*ϒout*), then the decision will be made to initiate the handover immediately to legacy LTE eNB macro cell. The network must be in a dual-connected mode to enable this process to be executed.(ii)If the SINR difference between serving mmWave gNB and target mmWave gNB is less than the predefined minimum HOM (*HO_min_*), then the maximum TTT value (*TTT_max_*) will be adapted, and the handover decision will be made once the TTT value expires.(iii)If the SINR difference between serving mmWave gNB and target mmWave gNB is higher than the predefined maximum HOM (*HO_max_*), then the minimum TTT value (*TTT_min_*) will be adapted, and the handover decision will be made once the TTT value expires.(iv)If the SINR difference between serving mmWave gNB and target mmWave gNB is between the minimum and maximum HOM (*HO_min_* < α < *HO_max_*), then the adaptive TTT interval (*α TTT*) will be selected based on Equation (6), and the handover decision will be made once the adaptive TTT value is expired.

(6)$$\propto_{TTT}=TTT_{max}-\left(Ď\right)\times \left(TTT_{max}-TTT_{min}\right)$$
where Ď is the deciding factor of the 5G mmWave channel condition, which is represented by the following:(7)$$Ď=\frac{\Delta_{SINR}-HO_{min}}{HO_{max}-HO_{min}}$$
where $\Delta_{SINR}$ is a SINR difference between the target and current 5G mmWave gNBs. When the $\Delta_{SINR}$ is equal to $HO_{min}$, then the deciding factor Ď value will be 0, and $\propto_{TTT}=TTT_{max}$, and on the contrary, if $\Delta_{SINR}$ is equal to $HO_{max}$ then the deciding factor Ď will become 1, and therefore $\propto_{TTT}=TTT_{min}$.

Thus, based on the proposed ATH mechanism, the TTT intervals are adaptively open for timely adjustment based on the instantaneous channel condition of the mmWave network.

The deciding factor, Ď, has a role of giving different weights (i.e., 0 < Ď < 1) for the uplink paths to adapt the TTT handover parameter in order to compromise the network performance. Algorithm 1 describes the process flow of the proposed ATH mechanism.
**Algorithm 1: ATH Mechanism**Initialize System Parameters (5G mmWave network)Start Channel Monitoring (SINR Measurement)*Generation* of gMRTs by 5G mmWave gNBs*Generation* of eMRTs by centralized eNBs***SINR*** Evaluation by centralized eNBs***if* the *SINR*** *serving* gNB > ***SINR*** *target* gNB *Handover Decision → False****elseif*** all gNBs outage  *Handover Decision → True*  *Initiate* Handover to centralized eNBs immediately***elseif SINR*** difference between *serving* and *target* gNBs < ***HOM_min_*** ***Output:*** **Adapt *TTT_max_******if TTT*** expired and none of *neighboring* gNBs have highest ***SINR***  *Handover Decision → True* *Initiate* Handover to target gNB***else*** *Handover Decision → False****elseif*** *SINR* difference between serving and target gNBs > ***HOM_max_*** ***Output: Adapt TTT_min_******if TTT*** expired and none of neighboring gNBs have highest ***SINR***  *Handover Decision → True* *Initiate* Handover to target gNB***else*** *Handover Decision → False****elseif*** *SINR* difference between serving and target gNBs is ***HOM_min_*** < α < ***HOM_max_*** ***Output:***
**Adapt α *TTT*** based on ***Equation* (*6*)*****if TTT*** expired and none of neighboring gNBs have highest ***SINR***  *Handover Decision _True* *Initiate* Handover to target gNB***else*** *Handover Decision → False****end******end******end******end***

On the other hand, as depicted in Figure 5, the HCPs (TTT and HOM) for the traditional handover mechanism will be fixed with the same settings (thresholds) at the base stations. Therefore, all UEs in the network are controlled by utilizing the same HCPs setting, leading to lower SINR adaptation, eventually causing the inefficient selection of a Modulation and Code Scheme (MCS). The fixed function also may lead to increased handover issues, such as RLF and HOPP, especially during UE mobility in highly dense environments. The issue becomes even more critical due to UEs’ mobility in a small coverage area of 5G mmWave gNB, which requires frequent handovers. Furthermore, implementing an adaptive TTT solution solely depends on the channel’s condition at a time, t. It means the implementation of a traditional handover is still possible for certain network environments, based on channel conditions, while the others may require an adaptive solution. A combination of a traditional and adaptive handover solution is essential and needs to be addressed in realizing 5G mmWave networks.

The timing diagram of ATH for a DC network is depicted in Figure 6. The DC framework is used in this work, allowing an uninterrupted connection between the UE, mmWave gNB, and LTE eNB points. Based on the diagram, the decision on the handover event will be made by a master LTE eNB once it acquires complete knowledge of the channel condition through eMRT. At this point, when a neighboring/target mmWave gNB has a better SINR value compared to the current gNB, the master eNB will initiate adaptive TTT intervals based on the algorithm proposed in Figure 2. Once the TTT interval has been initiated, the master eNB will hold the condition until the TTT interval expires and eventually triggers the handover to the target gNB by sending a secondary cell handover (SCH) request. The rest of the handshaking process will occur, as shown in Figure 6, until the handover is complete.

## 4. Simulation Framework

A typical urban grid network scenario is shown in Figure 7. It has a network area of 500 × 200 m, where multiple non-overlapping buildings are deployed with different sizes and heights. This is to randomize the channel dynamics and intermittency for the moving user.

Three mmWave eNBs are located at different coordinates, namely gNB 1 = (0; 100), gNB2 = (300; 100), and gNB3 = (200; 210), at a height of 5 m. A single UE is at coordinates (70; −10) at the beginning of the simulation. The UE will be moving through *x-axis* at a certain speed, *v* (km/h), until it arrives at a predefined end position. The simulation duration, *Tsim*, therefore depends on the UE speed, *v*, and is given by *T_sim_ = l_path_/v*.

The channel model and simulation parameters are defined in accordance with 3GPP Release 16 [33,34,35]. The 3GPP defined mmWave frequency with 28 GHz as a prominent candidate frequency band considered in a 5G system to meet the increasing demand for user data throughput. The simulation parameters are presented in Table 3.

In this work, we used a full stack of end-to-end mmWave modules that were presented by Mezzavilla in [6]. Since the Physical (PHY) and Medium Access Control (MAC) layers are scalable and extremely flexible, combining algorithms in these protocol stacks is simple. The stacks were interfaced and linked with the NS3 Long Term Evolution (LTE) module, and advanced architectural features such as DC were also presented. To help readers understand and further investigate the modules’ operations, a detailed explanation of the full-stack mmWave modules and their algorithms were provided in [6].

The mmWave channel conditions are highly unpredictable and intermittent. Since these signals are blocked by materials such as stone, concrete, and the human body, even a minor obstruction may induce a significant attenuation. All the earlier mentioned algorithms in the literature were designed to operate based on an ideal channel condition and not on the basis of a realistic mmWave channel that is highly intermittent and exposed to obstacles. In this work, we simulated our proposed ATH algorithm under unpredictable mmWave channel behavior that improves handover issues when implemented in a 5G network. As the signals deteriorate quickly in a 5G mmWave network, complex channel scanning is required to assist handover management. Therefore, to provide a realistic assessment of a 5G mmWave cellular network and to generate realistic dynamic models for link evaluation, a complex channel model, as presented in [6], is used. This channel model is based on an actual measurement at 28 GHz, and it is used to compute and estimate the SINR between UE and mmWave gNB by considering local blockage/obstacles and emulating the rapid variation of channel quality in a 5G mmWave scenario.

The algorithm was also tested under a 5G mmWave statistical channel model to represent a time-varying channel matrix that includes fading and the Doppler effect. The large-scale fading parameters are also defined during this phase to simulate the sudden change in the link quality. This is to realistically capture the blockage and intermittency events under the considered 5G mmWave bands. A detailed explanation of this simulation framework can be found in [6].

The existing handover algorithms were patched into NS3, along with all the supporting models, such as the channel path loss, transport agent, and propagation models. The DC cellular networks were implemented in compliance with 3GPP standards.

## 5. Performance Analysis

This section discusses the simulation results of the proposed ATH mechanism over traditional and other competitive handover mechanisms found in the literature. The ATH mechanism is evaluated using three different handover performance metrics: handover probability, latency, and PDCP throughput. The ATH algorithms are validated using a simulation with a dual-connectivity network, as explained in the previous session. For the first part of this section, the performance of the handover algorithm is assessed for different speed scenarios (30, 70, and 110) km/h and with different TTT handover parameters.

Figure 8 depicts the HOP with different mobile speed scenarios and TTT intervals. The HOP is gradually decreased with the increase of the UE speed in the given network scenario. This happens because, when the UE moves with high speed, it takes less time to exit the cells’ edge, eventually decreasing the HOP and vice versa. The shorter (i.e., 25 ms) TTT intervals (at both high and low speeds) increase the HOP dramatically; meanwhile, the longer (i.e., 200 ms) TTT intervals significantly reduce the HOP for the same speed scenarios. Even though longer TTT intervals decrease the HOP probability, this negatively affects the RLF rate, which is important to maintain the UE link connectivity with the base station. In summary, the TTT intervals have a direct relationship with the RLF and inverse relationship with the HOP. The findings in [32] also further support these claims, where the TTT intervals may cause a longer waiting time for handover decisions to be made, therefore increasing the RLF and decreasing the HOP. As observed in Figure 8, the highest HOP is seen when the TTT interval is 25 ms. This is because it requires a shorter waiting time for handover decisions to be made. This will lead to a higher HOP and eventually degrade the network’s performance. In contrast, the lowest HOP can be observed when the TTT intervals are high (i.e., 100 ms and 200 ms). As reported earlier, the longer TTT intervals require a longer waiting time to initiate handover, therefore leading to handover failure, and thus degrading the network performance, too. Through this finding, we can conclude that the TTT intervals play a vital role in preventing unnecessary and delayed handovers. Therefore, it needs to be tuned carefully by using the adaptation technique proposed in this work which is essential in the 5G mmWave network.

Figure 9 depicts the latency performance with different UE speeds and TTT intervals. As shown in the earlier graph (i.e., Figure 8), the latency is gradually decreased with the increase of the UE speed for the given network scenario. This happens because, when the UE moves with high speed, it takes a shorter time to reach the cells’ edge and eventually decreases the latency and vice versa. The highest latency is observed when TTT intervals are short (i.e., 25 ms); meanwhile, the lowest latency is observed when TTT intervals are high (i.e, 200 ms) for all speed scenarios. In general, the latency that is obtained by all TTT intervals, as observed in the graph, is still considered high where 5G communication is concerned. This is due to either the higher HOP that is caused by a shorter TTT or the higher RLF rate that is caused by a longer TTT or the combination of higher HOP and RLF that is caused by the moderate TTT intervals. For example, the latency (i.e., 7.17 ms) that is obtained at the UE speed of 30 km/h for TTT interval of 25 ms was caused by the higher HOP; meanwhile, the latency (i.e., 6.13 ms) that was obtained at the UE speed of 30 km/h for the TTT interval of 200 ms was caused by the higher RLF rate. Therefore, the TTT adaptation for handovers is essential to reduce the latency and boost the network performance in 5G communications.

In the simulations, the proposed ATH mechanism is compared with four competitive techniques. The following state-of-the-art algorithms were implemented for the performance comparison:[1]Dynamic HCP optimization mechanism proposed for LTE-A 5G Mobile Communication in [21]. In figure, this is denoted by Alhammadi HO.[2]Adaptive Time-to-Trigger Scheme proposed in [23]. In figure, this is denoted by Kim HO.[3]Traditional handover mechanism is implemented according to 3GPP, as defined in [33]. In figure, this is denoted by Traditional HO.[4]Best Connection (BC) representing the case when the UE is always connected to the mmWave gNB with highest SINR. In figure, this is denoted by BC HO.

In Figure 10, the performance comparison of handover latency over different beamforming configurations is depicted. It can be observed that the latency increases for different configurations of beamforming. This increase is caused by a less adaptation opportunity provided by other hybrid and analogue configurations compared to digital, therefore leading to rapid drops in SINR and consequent failures of the handover. The figure further shows that the proposed ATH mechanism enables the UE to optimally switch to the best mmWave gNB base station and eventually outperformed other competitive techniques. This significant performance is achieved by the fact that our proposed ATH mechanism optimizes the handover parameters, such as TTT, that are directly related to user’s connectivity. On the other hand, when the UE is attached to a certain base station for a longer time without a proper adaptation strategy, it will cause more latency in forwarding the packets to the new suboptimal cell once the handover is triggered.

Figure 11 demonstrates the comparative results of an average PDCP throughput between a proposed adaptive handover mechanism with other competitive techniques mentioned above. The average PDCP throughput is measured by sampling the received PDCP PDUs and summing the received packet sizes to obtain the total number of received bytes *B*(*t*). Then, the average PDCP throughput *S*(*t*) can be computed in bit/s as *S*(*t*) *= B*(*t*) *×* 8/*Ts*, where *Ts* is the sampling time. As shown in the graph, the proposed ATH mechanism is superior to all other techniques used for comparison in this work. This is due to the effectiveness of the proposed ATH algorithm in dealing with the mmWave channel behaviors and triggering handover during the crucial period of intermittency in networks. This significant performance is achieved by the fact that our proposed ATH mechanism optimizes the handover parameters, such as TTT, that are directly related to user’s connectivity. Furthermore, when the UE is attached to a serving base station for a longer time, it will cause more latency in forwarding the packets to the new suboptimal cell once the handover is triggered, eventually affecting the PDCP throughput. In short, the proposed algorithm achieves the best performance compared with the competitive techniques.

## 6. Conclusions

The 5G mmWave signals using the millimeter wave (mmWave) spectrums are highly vulnerable to blockage, causing the devices, or UEs, to suffer from connection failure. Furthermore, a channel condition that is highly intermittent with respect to user mobility and obstacles causes a fluctuation in signal strength and may cause a sudden outage. The conventional fixed and other optimization-based TTT handover mechanisms proposed in the literature do not work competently under these unpredictable channel behaviors and eventually lead to performance degradation in 5G mmWave networks. In this work, we proposed an adaptive handover mechanism known as ATH, which automatically adjusts the handover control parameters, such as TTT, based on the current state of channel condition measured by SINR. The UE directionally broadcasts SRS in a time-varying direction to enable 5G gNB stations to determine SINR for each angular space and fill it in the proposed table. Once a centralized LTE-Advanced station in a dual-connected network acquires a full directional knowledge of signal quality from all 5G gNB stations within the network, it will adaptively tune the TTT parameter based on the proposed adaptive algorithm and subsequently make a handover decision. The performance of the proposed handover mechanism was analyzed and evaluated in terms of handover probability, latency, and throughput by using Network Simulator 3 simulation. The comparative simulation result shows that the proposed ATH mechanism performs excellently when implemented in multi-Radio Access Technologies network scenarios and is effective for UEs that differ in moving speed and changing network environment with the presence of obstacles. The results also show that the adaptive performance is significantly improved compared to the conventional handovers and other enhancement techniques. This research will indicate the future implementation of a 4G/5G network, especially in an ultra-dense area, such as a metropolitan city, to support a massive number of subscribers and to fulfill the various traffic demand based on multiple services and applications of 5G technologies.

## Figures and Tables

**Figure 1 sensors-23-04357-f001:**
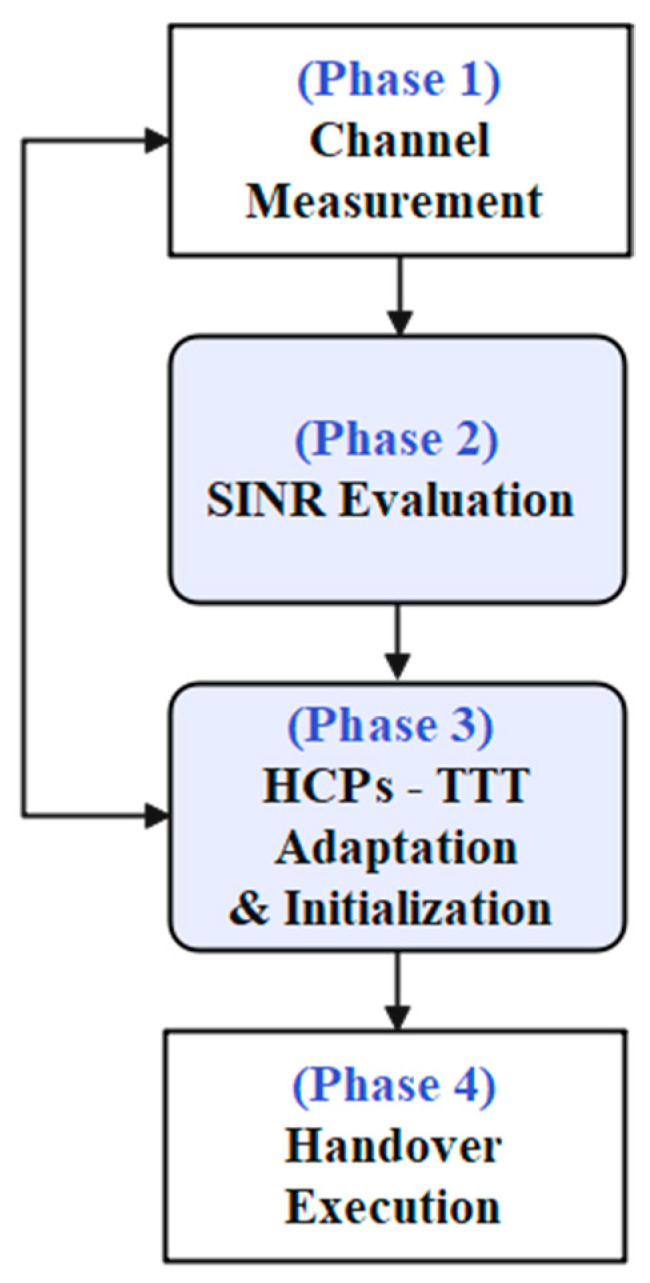
The diagram of proposed ATH mechanism.

**Figure 2 sensors-23-04357-f002:**
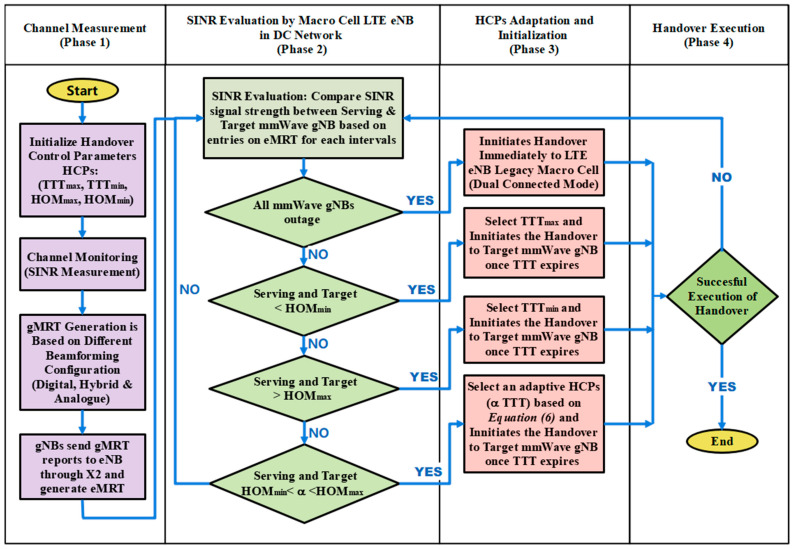
Design framework for an Adaptive TTT Handover (ATH) Mechanism.

**Figure 3 sensors-23-04357-f003:**
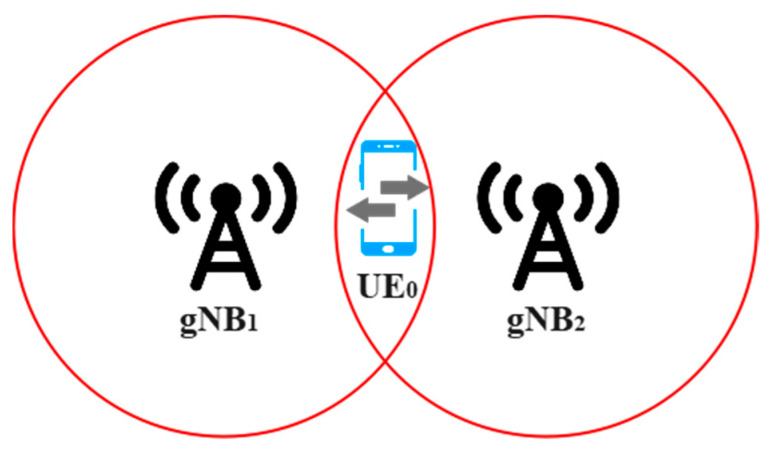
A 5G mmWave small cell network.

**Figure 4 sensors-23-04357-f004:**
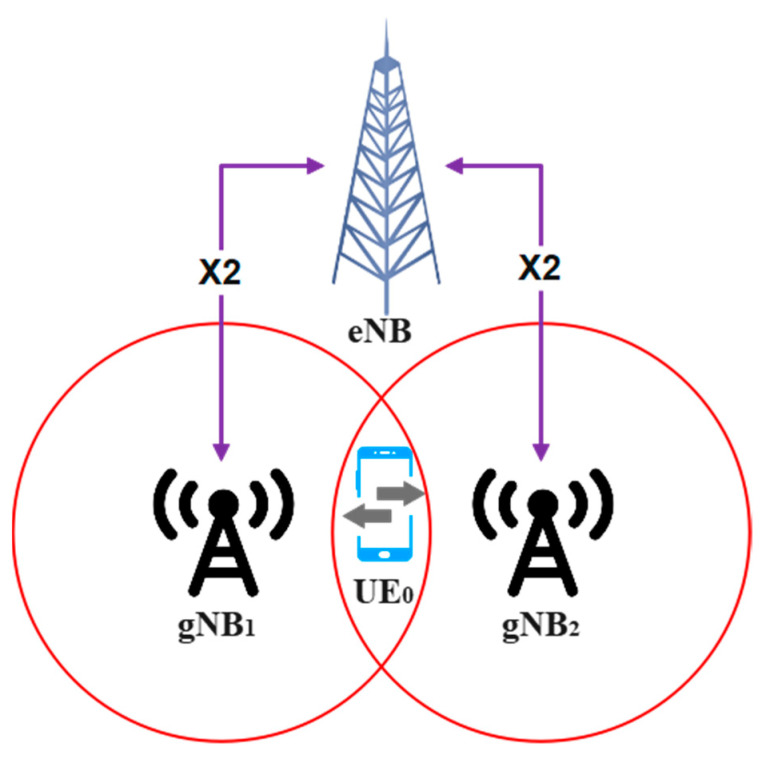
Dual-connected (LTE-5G mmWave) network.

**Figure 5 sensors-23-04357-f005:**
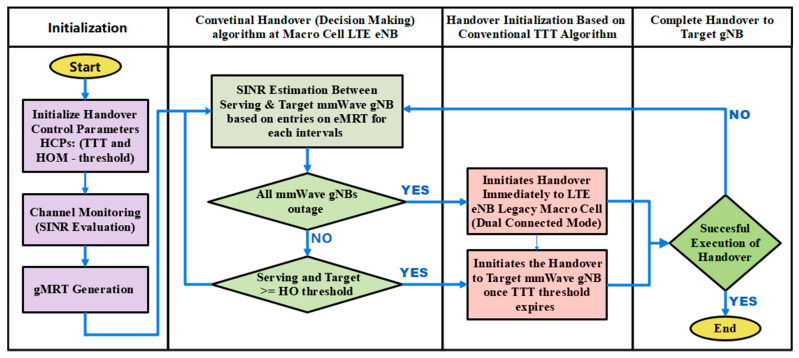
Conventional TTT handover mechanism.

**Figure 6 sensors-23-04357-f006:**
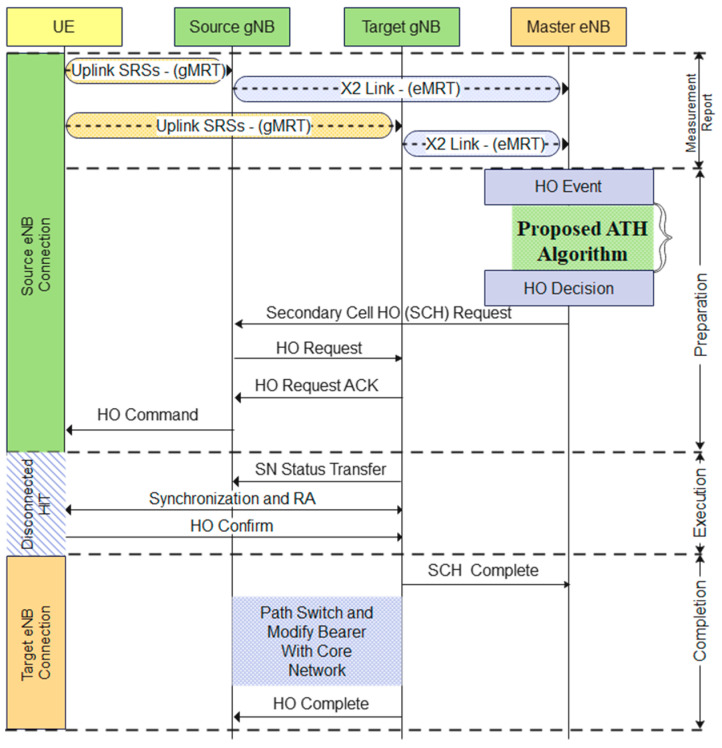
Timing diagram of the proposed ATH algorithm.

**Figure 7 sensors-23-04357-f007:**
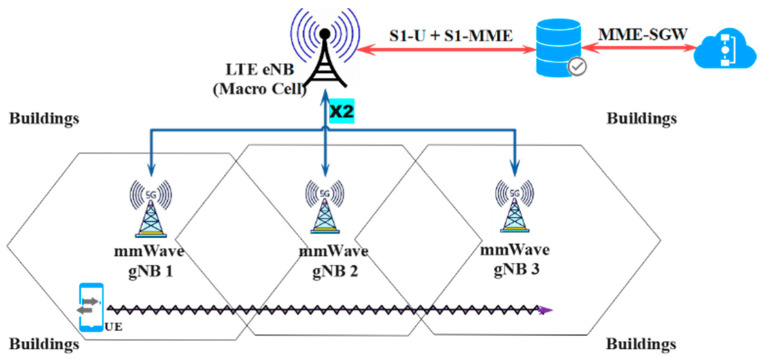
The reference simulation scenario for the DC network.

**Figure 8 sensors-23-04357-f008:**
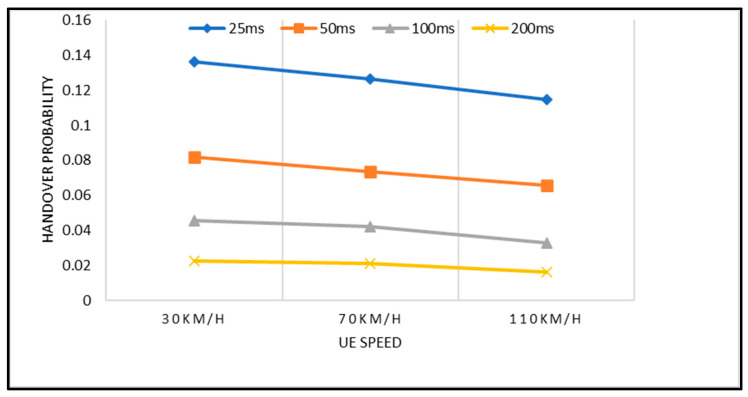
Handover probability with respect to UE Speed and TTT intervals.

**Figure 9 sensors-23-04357-f009:**
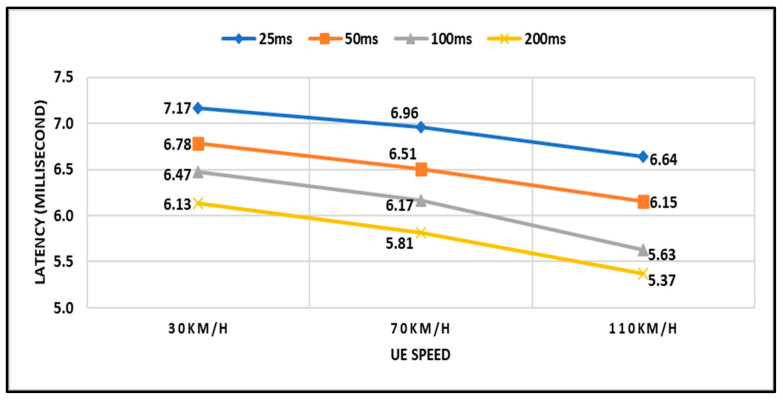
Latency with respect to UE Speed and TTT intervals.

**Figure 10 sensors-23-04357-f010:**
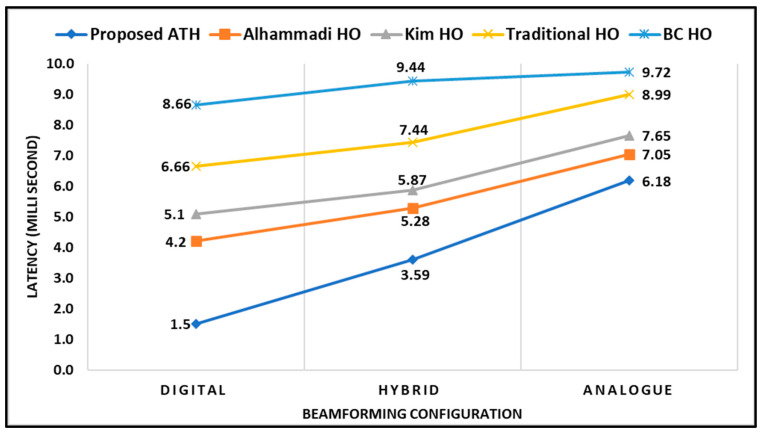
Latency (ms) with respect to Beamforming Configuration.

**Figure 11 sensors-23-04357-f011:**
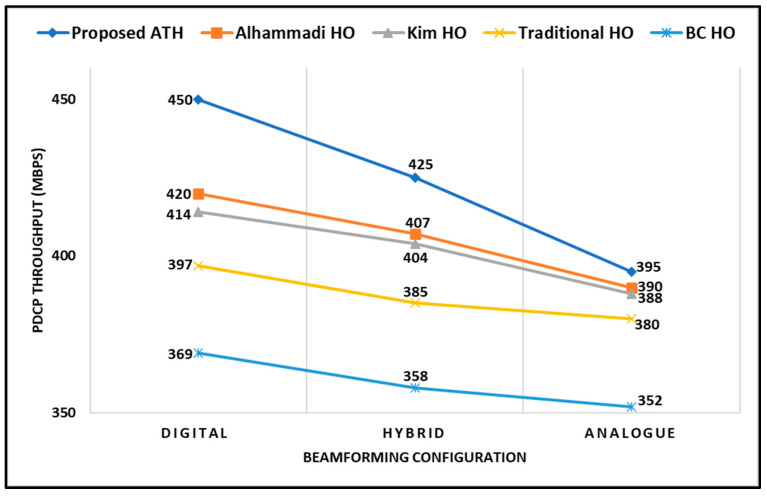
PDCP Throughput (Mbps) with respect to Beamforming Configuration.

**Table 1 sensors-23-04357-t001:** Summary of most recent related works.

Ref.	Issues	Solution	Technology	Parameters	KPIs
[15]	Unstable connection of UE due to user mobility	RHOT-FLC Technique	Beyond 5G Networks	TTT and HOM	HOP, HOF, HOPP, HOL, and HIT
[16]	Unnecessary HOs in Ultra Dense Networks	FLDHDT Technique	NSA (4G–5G)	TTT and HOM	HOP, HOPP, and throughput
[17]	Poor HO management in the high mobility of a dense network	LIM*2* Technique	Beyond 5G Networks	TTT and HOM	Throughput, packet loss rate, and RLF
[18]	Increases of interference and HOs in HetNets	ATO Algorithm	LTE-A and 5G HetNets	TTT and HOM	Call drop rate, HO delay, and HIT
[19]	Rapid fluctuations in the signal strength caused unnecessary HOs	AHOM-NWF Algorithm	LTE-A Network	HOM	SINR, cell edge SE, and outage probability
[20]	Mobility issues in urban environments with 5G small cells	IDHPO-AWF Algorithm	5G Network	RSSI and SNR	Throughput, delay, and packet loss rate
[21]	Mobility issues in mmWave network that effect HO performance	Dynamic HCPs	LTE-A and mmWave HetNets	TTT and HOM	HOP and RLF
[22]	Increase of latency, handover failures, and frequent handover in UDNs	Entropy-based weighting decision-making method	5G Network	Bandwidth, user density, and SINR	Delay, block ratio, HOF, and throughput
[23]	Inefficient HCPs setting due to UE’s velocity	Adaptive TTT Technique	LTE Network	TTT and HOM	HOP, DLF, and HOPP
[25]	Increase of frequent handovers and overheads in UDNs	Reinforcement Learning Method	mmWave Network	Beamforming	HOP and trajectory length
[27]	Unstable connection due to mmWave signals that are highly susceptible to blockage and extremely intermittent	A novel uplink control signaling Technique for DC	Dual Connected (LTE—mmWave)	SINR, Beamforming, and TTT	Handover events, latency, PDCP throughput, and RRC traffic

**Table 2 sensors-23-04357-t002:** The 5G signal quality parameters for different RSSI (RSRP, RSRQ, and SINR).

		RSRP (dBm)	RSRQ (dBm)	SINR (dB)
Radio Frequency (RF) Condition	Excellent	>−80	>−10	>20
	Good	−80 to −90	−10 to −15	13 to 20
	Mid Cell	−91 to −100	−14 to −20	0 to 12
	Cell Edge	<−100	<−20	<0

**Table 3 sensors-23-04357-t003:** Simulation parameters.

Parameters	Values
mmWave, *fc*	28 GHz
mmWave P_TX_	30 dBm
LTE, *fc*	2.1 GHz
Noise Figure (NF)	5 dB
SINR_threshold_ (ϒout)	−5 dB
LTE_MIMOantenna_	8 × 8
UE_MIMOantenna_	4 × 4
SRS_duration_	10 µs
SRS_period_	200 µs
RLC_buffersize_	10 MB
X2_delay_	1 ms
MME_delay_	10 ms
Payload_size_	1024 byte
TTT_max_	150 ms
TTT_min_	25 ms
HO_max_	8 dB
HO_min_	3 dB
Sampling_time_, *T_s_*	5 ms
UE*_speed_* (υ)	20 km/h

## Data Availability

Not applicable.

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
