# Peer review of "An Adaptive TTT Handover (ATH) Mechanism for Dual Connectivity (5G mmWave—LTE Advanced) during Unpredictable Wireless Channel Behavior"

_sensors, 2023, doi:10.3390/s23094357_

Round 1

Reviewer 1 Report (Previous Reviewer 1)

The second revision of the paper has now been improved. But I still have concern regarding the novelty of the paper.

This paper is essentially using adaptive Handover Control Parameter (HCP), i.e Time to Triger (TTT) and Hand Over Margin (HOM) to improve the handover performance under an unpredictable (intermittent) channel behaviour.

Since a dynamic as well as adaptive HCP has been previously proposed in [21] and [23], prior to accepting the paper for publication, can authors explicitly explain the difference between the adaptive HCP method proposed in [21] and [23] with that of proposed in this paper ?

Author Response

REVIEW 1

First of all, we would like to thank the reviewer who has provided excellent remarks to improve the paper. We agree with the majority of the comments. Please find our feedback of improvement in blue.

Reviewer:

Since a dynamic as well as adaptive HCP has been previously proposed in [21] and [23], prior to accepting the paper for publication, can authors explicitly explain the difference between the adaptive HCP method proposed in [21] and [23] with that of proposed in this paper?

Our reply: -

The novelty of our proposed Adaptive TTT Handover (ATH) mechanism as presented in earlier version can be found in from page 3 Line 106-122. The ATH is a modification from previous works as published by [19, 21 and 27], based on SINR on instantaneous channel conditions that can occurred in Beyond 5G networks coexistence with LTE. The handover decision making algorithm based on TTT adaptation as presented by using Figure 2 (Page 9 line 383) and adaptive weight function formula (Page 12 – Equation 6 & 7) is a new technique/method that proposed in this work.

Furthermore, the adaptive HO algorithm is evaluated under unpredictable channel behavior and was compared not only with the traditional method [33], but also with other competitive method in literature [21] and [23] as a benchmark improvement method. We also proved that the proposed ATH mechanism significantly outperformed other handover mechanisms as shown in result section (Page 17 – 20). It shows the effectiveness of the proposed ATH algorithm as presented in (Page 13 Line 550) dealing with mmWave channel behavior to make an appropriate handover decision.

The analysis result (Page 17 – 20) shows that our proposed ATH mechanism outperformed the technique proposed by Alhammadi HO [21] and Kim [23]. Therefore, the adaptive TTT method that proposed by both Alhammadi HO [21] and Kim [23] by only exploiting predefined values based on detected events (without considering the channel condition and interference) not practical and suitable for 5G channel behavior that highly intermittent and susceptible to even minor blockage.

We can claim that the effectiveness of our ATH mechanism in adapting TTT parameter much more accurate and reliable compared to the simple TTT adaptation method proposed by Alhammadi HO [21] and Kim HO [23]. This claim strengthening originality of our contribution with respect to Alhammadi HO [21] and Kim HO [23]

The (attached) table summarized in detail the originality of our contribution with respect to Alhammadi HO [21] and Kim HO [23]. 

Reviewer 2 Report (New Reviewer)

The overall impression from the presented submission is quite satisfactory. The text is well-written and well-formatted. The authors' logic and all the derivations are clear. The obtained results are somewhat interesting. But there are several issues with the submission.

The text abounds with different abbreviations (which is normal for a network research), but some of them are not expanded at all (for example, see figures, and some places in the text). It is hard to read the text without a proper glossary. It is advised to add an abbreviation table at the beginning of the submission (or at the end).  

There are only few equations, but they are formatted poorly. Please, remember that the equations are the parts of the sentence, so they must be followed and preceded with punctuation marks.

In Eq. (1), the authors multiply the denominator by “noise power”?! This is clearly wrong! Moreover, what is “Wtot” and “x”?

In Eq. (2), what is the need for summation sign, since there is only one term?

The lines 421-440 must be rewritten, since it is simply one sentence with variable’s itemization, nothing more.

In Eq. (5), “d” is explained as “distance”, which is clearly wrong, since it is used under the “log” operator. Usually, it is “a distance normalized to some reference displacement”. So what was the “reference displacement” used in simulation?

Table 2 presents not the “relationship between RSRP, RSRQ and SINR”, but rather the relationship between their performance/values for different conditions.

In Fig. 1, there is a double-sided arrow, which means that not only Phase 3 stage is adapted according to the results on Phase 1 stage, but also vis versa. But nothing is said about that back loop in the text.

In lines 705-711, the reasoning behind the fact that the increase of speed decreases handover probability is unclear and not convincing. It is not hard to present a counterexample. So this explanation must be elaborated and strengthened.

The authors omitted the results of PDCP throughput and latency for different TTT and UE speed. This must be provided in the revision.

Author Response

REVIEW 2

First of all, we would like to thank the reviewer who has provided excellent remarks to improve the paper. We agree with the majority of the comments. Please find our feedback of improvement in blue.

Reviewer:

  1. The text abounds with different abbreviations (which is normal for a network research), but some of them are not expanded at all (for example, see figures, and some places in the text). It is hard to read the text without a proper glossary. It is advised to add an abbreviation table at the beginning of the submission (or at the end).  

Our reply: -

We have revised the and include the abbreviations list at the end of Manuscript. The abbreviation list can be found at page 21

  1. There are only few equations, but they are formatted poorly. Please, remember that the equations are the parts of the sentence, so they must be followed and preceded with punctuation marks.

Our reply: -

We have revised this, and all amendments had made on formulas and properly formatted. This can be found from page 10 (line 399) to page 11 (line 445)

  1. In Eq. (1), the authors multiply the denominator by “noise power”?! This is clearly wrong! Moreover, what is “Wtot” and “x”?

Our reply: -

Its actually not multiple but add the denominator by “noise power. We have missed out (+) operator in the formula at denominator. The revision has made on the formula.  This can be found at page 10 (line 399)

  1. In Eq. (2), what is the need for summation sign, since there is only one term?

Our reply: -

We have revised this and removed the summation sign as suggested my reviewer. This can be found at page 10 (line 418)

  1. The lines 421-440 must be rewritten, since it is simply one sentence with variable’s itemization, nothing more.

Our reply: -

The revision has made on this sentence as suggested my reviewer. This can be found from page 10 (line 399) to page 11 (line 445)

  1. In Eq. (5), “d” is explained as “distance”, which is clearly wrong, since it is used under the “log” operator. Usually, it is “a distance normalized to some reference displacement”. So what was the “reference displacement” used in simulation?

Our reply: -

This is log distance pathloss model where average received signal power decreases logarithmically with distance. We have revised the formula as suggested by the reviewer and it can be found from page 10 (line 428) to (line 436)

  1. Table 2 presents not the “relationship between RSRP, RSRQ and SINR”, but rather the relationship between their performance/values for different conditions.

Our reply: -

We have revised the term for the table 2 and labeled it correctly according to the comment received. This can be found at page 9 (line 372)

  1. In Fig. 1, there is a double-sided arrow, which means that not only Phase 3 stage is adapted according to the results on Phase 1 stage, but also vis versa. But nothing is said about that back loop in the text.

Our reply: -

We explained regarding of the back loop it in our previous version. However, as suggested by the reviewer, we have improved the sentence to look more comprehensive. This can be found from page 8 (line 329) to (line 341)

  1. In lines 705-711, the reasoning behind the fact that the increase of speed decreases handover probability is unclear and not convincing. It is not hard to present a counterexample. So this explanation must be elaborated and strengthened.

Our reply: -

We have revised the whole sentence as per suggested by the reviewer and strengthened the statement with more elaboration.  This can be found from page 17 (line 679) to (line 699)

  1. The authors omitted the results of PDCP throughput /latency for different TTT and UE speed. This must be provided in the revision.

Our reply: -

We have revised and included additional result at Page 18 (Line 703) as per requested by the reviewer.

Reviewer 3 Report (New Reviewer)

No comments in this version.

Author Response

No comments in this version

Round 2

Reviewer 2 Report (New Reviewer)

I thank the authors for correct responding to my questions, sufficient extension and improvement of the submission. Most of my issues has be resolved. I have no further objections from accepting it for publication.

This manuscript is a resubmission of an earlier submission. The following is a list of the peer review reports and author responses from that submission.

Round 1

Reviewer 1 Report

Title : An Adaptive Handover Mechanism for Dual Connectivity (5G mmWave – LTE-Advanced) during  Unpredictable Wireless Channel Behaviour

Authors : Vigneswara Rao Gannapathy,  Rosdiadee Nordin, Nor Fadzilah Abdullah, Asma Abu-Samah, Mahamod Ismail.

This article proposes an adaptive handover mechanism, which deals with unpredictable wireless channel behaviour at dual-connected (5G mmWave - LTE) bands. The algorithm is developed to automatically adjust the handover control parameters such as time-to-trigger (TTT) and handover margin (HOM) based on the current state of Signal to Interference Noise Ratio (SINR) of the link. The unpredictable wireless channel is represented by mmWave statistical channel model that is used to generate time varying channel matrix that includes fading and doppler effect parameters. Then, the performance of the proposed handover mechanism is analyzed by using NS3 simulation tool. The proposed method is evaluated in term of throughput and latency and compared with the existing dual connectivity-based handover mechanisms.

Review :

The quality of manuscript is inadequate for a journal publication. The authors should highlight the new novelty of the proposed hand over mechanism, instead of adjusting the hand over control parameters and listing the simulation results of NS-3 simulation tool without theoretical justification of the hand over model in this paper. The authors propose automatic adjustment of the hand over parameters, i.e. time-to-trigger (TTT) and hand over margin (HOM) based on the current state of signal-to-interference plus noise ratio (SINR) of the link.  It seems that the paper neither contain enough novel idea, nor contain any distinct and significant contributions to new knowledge in hand over algorithm compared to previously reported papers (see additional comments below). The author needs to clearly explain what new hand over method in this design are different from the previously published hand over algorithm. Unfortunately, a novel thing or useful for reader is not found. The paper also has a very limited proposed work, so that the scope of the paper is less comprehensive. Therefore, the paper would be more suitable for a conference publication, but would not be considered appropriate to publish in the journal.

Additional comments :

Authors need to introduce the difference between hand over mechanism in dual connectivity network with that of conventional single connectivity network, so the motivation behind this proposed work can be strongly justified.

Authors need to see the previous work published by :

1.       Jose et.al., “Adaptive TTT Scheme to Optimize Handover in High Speed Environment,”  International Journal Of Scientific & Engineering Research, Volume 7, Issue 7, July-2016.

2.       Kim et.al, “Adaptive Time-to-Trigger Scheme for Optimizing LTE Handover,” International Journal  of Control and Automation Vol.7, No.4 (2014), pp.35-44.

Authors need to explain why signal-to-interference plus noise ratio (SINR) is proposed as hand over control parameter, rather than using received signal power parameter (RSPP) that is much easier to obtain and readily available in the network. SINR is more accurate as control parameter for highly interference limited network, but it is very difficult and very costly to obtain.

The detail of simulation framework, either the framework of the proposed design, and also the simulation framework of existing hand over mechanism, need to be clearly explained to realistically capture and compare the incremental improvement of the proposed design with the existing hand over mechanism.

The simulation procedures need to be described and formally validated that the simulation is conducted correctly, and the results can be validated by theoretical/mathematical expression of hand over algorithm employed in the simulation. The simulation results shown in Fig. 4 and Fig 5. are difficult to be judged by the hand over model developed in this paper.

Authors claim the proposed scheme outperform the conventional method of handover method, but author does not provide performance comparison between the proposed and the conventional hand over schemes.

Authors show the performance in terms of latency and throughput. But authors only read out the curves obtained from simulation (the readers can read the curves by themselves), without explaining the effect of control parameter on the performance of latency, i.e. why increasing hand over margin (HOM) and the UE speed of mobility increases the latency (Fig.4). Authors need to develop theoretical/mathematical models to express the fading (signal fluctuation) and Doppler, so that the effect adjusting the hand over margin (HOM) and time-to-trigger (TTT) depicted in equation (1) on the latency performance can be justified.

Beamforming configuration was not evaluated in the simulation, but it appears as performance indicator in Fig. 5, so that authors fail to explain the effect of beamforming configuration (digital, hybrid, and analogue) on the performance of latency (why hybrid beamforming configuration has larger latency than digital beamforming and also why  analogue beamforming has larger latency than hybrid beamforming). Again, authors only read out the curves in Fig. 5, instead of discussing the reason why ?

Language:

Check the correctness of English language, e.g. :

As shown in the graph, in comparison with the latency, its proven the proposed adaptive TTT handover mechanism is outperformed the dynamic [12] and fixed TTT for all three-beamforming configurations known as digital, hybrid and analogue.

Recommendation :

Reject for this journal, suggested to submit to conference/letter publication.

Author Response

Please find attached the letter to the reviewers. 

We were happy to see such constructive comments.
We hope the corrections have addressed the proposed improvements.

Reviewer 2 Report

This paper presents an adaptive handover mechanism for dual connectivity (5G mmWave – LTE-Advanced) during unpredictable wireless channel behavior. I have the following concerns:

1. References should be given serially starting from [1].

2. 5G spectrum should be given. In 5G, there is also Sub-6GHz networks for rural and suburban areas [i]. Did you consider that?

[i]  Shah, A. F. M. S. A Survey From 1G to 5G Including the Advent of 6G: Architectures, Multiple Access Techniques, and Emerging Technologies. In Proceedings of the IEEE 12th Annual Computing and Communication Workshop and Conference (CCWC), Las Vegas, NV, USA, 26-29 January 2022, pp. 1117-1123.

3. Abbreviation should be given once in the beginning, then the short form can be used. Do not put the third bracket when it is not in short form. Check page 2, line 86, TTT.

4. At the end of section 1 (introduction), give a paragraph describing how the paper is organized.

5. In Figure 1, SINR estimation and evaluation are mentioned but how will you estimate and evaluate SINR?

6. In section 5, a performance comparison with existing works should be given. 

7. Please proofread the manuscript. There are a lot of typos.

Author Response

Thank you for the proposed improvements. We really appreciate the comments. Please find attached the response letter. We hope we have done justice to the correction process. 

Round 2

Reviewer 1 Report

After careful evaluation on author's respond to the reviewer's comments and also read carefully the revised version of the manuscript, unfortunately the paper is not sufficient for publication in a Journal, for the following reasons:

1.   The paper does not have significant contribution to new knowledge on the hand over method, rather than adjusting the hand over parameters, i.e. Time to Triger (TTT) and Hand Over Margin (HOM). Therefore, it doesn't have significant novelty.

2. Author's response to the reviewer's comments is unsatisfactory. For example, authors fail to explain the difference between the proposed TTT adjustment in their paper with that of published in: Kim et.al, “Adaptive Time-to-Trigger Scheme for Optimizing LTE Handover,” International Journal of Control and Automation Vol.7, No.4 (2014), pp.35-44. Authors also fail to properly explain why costly SINR is used as control parameter, instead of using simple and readily available signal strength.

3. The scope of the manuscript is too limited for a journal publication. It is more suitable for a conference paper, rather than for a journal publications.

Recommendation:

The paper cannot be accepted for publication

Reviewer 2 Report

After the revision, the paper is improved. Still, I have the following concerns:

1. In Figure 1, SINR estimation and evaluation are mentioned but how will you estimate and evaluate

SINR? The previous reply is not clear. You cited [30]. Which equations are used?

2. There are works on adaptive handover mechanisms. A performance comparison with existing works should be given. It can be a quantitative analysis.

3. What do you mean by conventional time-to-trigger (TTT)? In the result section, you used it for comparison. Can you cite any article for conventional TTT?

4. Few more recent articles on 5G and beyond communications can be included to improve the literature survey.

i. Mobility Management in 5G and Beyond: A Novel Smart Handover with Adaptive Time-to-Trigger and Hysteresis Margin. IEEE Transactions on Mobile Computing, 2022.

ii. Architecture of Emergency Communication Systems in Disasters through UAVs in 5G and Beyond. Drones, 2023.

iii. A Survey From 1G to 5G Including the Advent of 6G: Architectures, Multiple Access Techniques, and Emerging Technologies. In Proceedings of the IEEE 12th Annual Computing and Communication Workshop and Conference, Las Vegas, USA, 26-29 January 2022.

iv. Multi-criteria handover management using entropy‐based SAW method for SDN-based 5G small cells. Wireless Networks, 2021.